# Bioinformatics Strategies to Identify Shared Molecular Biomarkers That Link Ischemic Stroke and Moyamoya Disease with Glioblastoma

**DOI:** 10.3390/pharmaceutics14081573

**Published:** 2022-07-28

**Authors:** Md Khairul Islam, Md Rakibul Islam, Md Habibur Rahman, Md Zahidul Islam, Md Al Amin, Kazi Rejvee Ahmed, Md Ataur Rahman, Mohammad Ali Moni, Bonglee Kim

**Affiliations:** 1Department of Information & Communication Technology, Islamic University, Kushtia 7003, Bangladesh; mdkito51@gmail.com (M.K.I.); mrpislam@gmail.com (M.R.I.); zahidimage@gmail.com (M.Z.I.); 2Department of Computer Science & Engineering, Islamic University, Kushtia 7003, Bangladesh; habib@iu.ac.bd; 3Department of Computer Science & Engineering, Prime University, Dhaka 1216, Bangladesh; ahmedalamin2357@gmail.com; 4Department of Pathology, College of Korean Medicine, Kyung Hee University, Hoegidong Dongdaemungu, Seoul 02447, Korea; kazirejveeahmed@gmail.com; 5Korean Medicine-Based Drug Repositioning Cancer Research Center, College of Korean Medicine, Kyung Hee University, Seoul 02447, Korea; 6School of Health and Rehabilitation Sciences, Faculty of Health and Behavioural Sciences, The University of Queensland, St Lucia, QLD 4072, Australia

**Keywords:** glioblastoma, ischemic stroke, moyamoya, bioinformatics, association, GSEA, pathway, orthology

## Abstract

Expanding data suggest that glioblastoma is accountable for the growing prevalence of various forms of stroke formation, such as ischemic stroke and moyamoya disease. However, the underlying deterministic details are still unspecified. Bioinformatics approaches are designed to investigate the relationships between two pathogens as well as fill this study void. Glioblastoma is a form of cancer that typically occurs in the brain or spinal cord and is highly destructive. A stroke occurs when a brain region starts to lose blood circulation and prevents functioning. Moyamoya disorder is a recurrent and recurring arterial disorder of the brain. To begin, adequate gene expression datasets on glioblastoma, ischemic stroke, and moyamoya disease were gathered from various repositories. Then, the association between glioblastoma, ischemic stroke, and moyamoya was established using the existing pipelines. The framework was developed as a generalized workflow to allow for the aggregation of transcriptomic gene expression across specific tissue; Gene Ontology (GO) and biological pathway, as well as the validation of such data, are carried out using enrichment studies such as protein–protein interaction and gold benchmark databases. The results contribute to a more profound knowledge of the disease mechanisms and unveil the projected correlations among the diseases.

## 1. Introduction

Glioblastoma, generally regarded as glioblastoma-multiforme (GBM), is the most deadly form of cancer in the brain region throughout the world [1]. Percival Bailey and Harvey Cushing introduced the name glioblastoma multiforme in 1926, emphasizing the hypothesis that the cancerous cells arise from Gila, fundamental drug precursors (glioblasts). Additionally, it is an utterly volatile presentation caused by necrosis, hemorrhage, and cysts (multiform) [2]. GBM has vague signs or symptoms initially. Headaches, mood changes, fatigue, and symbols close to those of a stroke are all possible symptoms [3]. Symptoms sometimes escalate quickly, leading to unconsciousness [4]. Recent studies suggest that astrocytes, brain stem cells, and oligodendrocyte progenitor cells could all represent the disease’s biological origin cell [5,6]. Glioblastoma stem cells have already been observed in patients with GBM, exhibiting features similar to progenitor cells. Their involvement, along with the dispersed form of glioblastomas, makes surgical removal impossible and is thus thought to be a potential source of resistance to medical therapies and a strong recurrence risk [7]. The disease affects around three out of every 100,000 people annually, but the rate can be even higher in certain areas [8]. Glioma has the highest mortality rate of the different forms of brain tumors, is an unusual curable shape, is immune to chemotherapy and radiotherapy, and has a poor prognosis [9].

Ischemic stroke victims are more likely to acquire a brain tumor, most often a glioma, due to the ridiculous effects of ischemia as well as the effects of hypoxia on a cell’s functional and metabolic state [10]. When blood circulation to a portion of the brain is eventually cut off, an ischemic stroke occurs, which leads to the loss of neurological control. Ischemia may also occur when blood circulation to a particular part of the brain is inadequate to satisfy physiological demands [11,12]. Thus, this causes a lack of oxygen in the brain (cerebral hypoxia) and, as a result, brain tissue dies (cerebral infarction/ischemic stroke) [13]. On the other hand, proliferating cell density, metastasis, and a general prothrombotic propensity correlated with tumors raise the likelihood of ischemic infarctions [9]. Several theories have been suggested to explain why ischemic infarction and brain tumors, especially glioma, occur together. The most prevalent mechanism highlights how both situations are prone to hypoxia [14]. Cerebral ischemia, for example, induces blood flow congestion and predisposes to hypoxia [15]. At the same time, a quickly increasing malignant mass has a hypoxic heart owing to the intensified need for oxygen from rapidly dividing cells [16]. Other possible pathways in the interplay between the two systems have been suggested by researchers, including astrocyte-activation [17], angiogenesis, reactive-gliosis [18], and various modifications in the tumor microhabitat [19]; some of which are primarily caused by cerebral ischemia as a result of quick glioma growth [9]. Moreover, frequent removal surgery (operation of any tissue or part of an organ) to treat gliomas raises the possibility of ischemic injury [20].

Moyamoya disease is a form of arterial occlusive disease that most frequently damages the brain’s carotid arteries. More specifically, carotid arteries narrow down or become blocked in the brain region, limiting blood supply to the brain [21]. It is identified by the angiographic characteristics of bilateral central carotid artery stenosis and unwanted expansion of the favoring veins in the brain’s center [22]. While the cause of primary moyamoya disease is unclear, moyamoya disease may be brought about by a number of different pathogenic reasons. Internal carotid arteries might become blocked as a result of intracranial basal tumors or radiotherapy, resulting in the formation of moyamoya-type vessels [23,24]. Hence, this is related to “leptomeningeal artery end-to-end anastomoses”, “transdural anastomoses”, and ’telangiectatic collaterals”, which are the most popular inside and outside areas of the basal ganglia [25]. In recent research studies, the relationship between glioblastoma and mm has been described [25,26].

In conclusion, there is convincing evidence that GBM, I. stroke, and mm have pathologically and medically significant connections, although this connection has not been thoroughly investigated. Since the etiology of GBM, I. stroke, and mm are complicated, and their risk factors differ in specific ways, the underlying relationship’s in biological aspects, and molecular mechanisms are still unknown. Despite their strong therapeutic relevance, GBM, I. stroke, and mm are very complicated disorders in terms of their clinical manifestations, making them challenging to analyze using traditional hypothesis-driven endocrinology analysis. Furthermore, there is still a scarcity of bioinformatics research on the issues discussed. The objective of this project was to find certain connections among diseases since knowing the existence of these connections could provide valuable insights into the diseases’ mechanisms. Therefore, this prompted us to design a bioinformatics framework to identify the essence of the interaction, such as gene expression and dysregulation, signaling pathways, and protein–protein interactions analyzed from disease-affected tissues. Results were then validated using experimentally validated gold benchmark databases and literature such as DisGeNET, db-GaP, and Rare-Diseases-AutoRIF.

## 2. Substances, Procedures, and Methods

### 2.1. Collected Datasets

The datasets included for the analysis were derived from the National Center for Biotechnology Information (NCBI), a well-known Gene Expression Omnibus (GEO) database. Each disease’s query returns a series of datasets. If the dataset is obtained from a non-human species and does not meet two criteria for each group, such as control samples (healthy) and case samples (patients), it is not preferred for our study. Additionally, we discarded repeated datasets, unfavorable formatting, or insignificant experimental emphasis. We also excluded datasets with sample sizes smaller than our preselected cutoff sample size of three for each group. Linear regression is used to analyze the transcriptomic differential expression of the selected GEO datasets, and a linear model may have appropriate analytical strength when the sample size for either the healthy or the patient is three, or higher than three [27]. In addition, we concentrated on a particular cell or tissue type in light of its influence on the course of a disease. This method resulted in selecting three strongly important datasets for glioblastoma, I. stroke, and moyamoya (mm) as well as suitable for the analysis. The datasets for glioblastoma and ischemic stroke are RNA-seq, and the dataset for moyamoya disease is micro-array. As no RNA-seq datasets met our requirements, we used the microarray dataset for moyamoya. We looked for datasets with the lowest amount of biases and distortion for this study. For the analysis, we selected transcriptome RNA-seq/microarray datasets of human participants with the accession numbers GSE106804, GSE56267, and GSE131293, which included both healthy and diseased patients.

The glioblastoma dataset (GSE106804) included gene expression data from the Extracellular Vesicle of 13 glioblastoma patients and 6 healthy controls [28]. GBM is constantly in contact with its underlying tumor microenvironment (TME). Extracellular vesicle has a significant effect on the GBM tumor microenvironment, paving the path for the development of GBM [29,30]. Hence, we selected the dataset. The I. stroke (GSE56267) dataset included gene expression evidence from the cortical tissue of seven I. stroke patients and six healthy controls, whereas cortical neurons depict important intact genome information regarding I. stroke patients [31]. The moyamoya dataset (GSE131293), the only microarray data, included gene expression results from three patients and three stable controls’ neural crest stem cells [32].

### 2.2. Preprocessing and Distinction of Differentially Expressed Genes

As mentioned earlier, the datasets were collected from NCBI. We performed differential expression analysis to detect the genes that are noticeably expressed in patients’ samples compared to healthy samples. We performed differential expression analysis (DEA) of RNA-seq raw count data using DESeq2, an R package. The internal normalization technique was carried out using DESeq2 and determined the geometric mean of every gene across all samples. Then, the negative binomial distribution, a linear model, was calculated for each gene, considering variability among samples. Finally, notable genes were filtered using the Wald test and we automatically removed low-expressed/outlier genes using Cook’s distance [33]. For microarray data, we used Limma, also a linear model, for DEA, which performed a *t*-test to find the importance of every gene over samples [34]. The code for DEA was implemented in R and can be accessed through our Github repository: https://github.com/hiddenntreasure/glioblastoma, accessed on 11 July 2022.

We used the Z-score transformation ($Z_{mn}$) for each disease phenotype to make the gene expression data more comparable. The equation for this transformation is
(1)$$Z_{mn}=\frac{g_{mn}-\bar{X}}{\sigma_{m}}$$
where $\sigma_{m}$ indicates standard deviation and $g_{mn}$ suggests the magnitude of the gene (*m*) in the sample (*n*). Thus, this allows us to directly measure the expression of genes across samples and types of cells from various disorders.

We discarded the genes with missing or null values. Two parameters are deployed to derive the most significant/biomarker genes accountable for the emergence of a disease. First, the *p*-value should be less than 0.05; secondly, the absolute value of the log2-fold change is either 1 or greater/less than 1. Genes with a logFC greater than one are highly expressed compared to the other genes and are known as upregulated (up-reg) genes, whereas downregulated (down-reg) genes are lower expressed in contrast to gene expression arrays and logFC is less than 1. We have several significant genes for each disease that are differentially dysregulated and significantly liable for developing a disease. Then, we identified shared genes between a pair: glioblastoma and I. stroke, as well as glioblastoma and moyamoya.

The prevalent genes in these two pairs of diseases were then used to build a gene-disease network (GDN), and different neighbors were found using Jaccard coefficient methods [35], which is the co-occurrence score. In contrast, the edge (connection among genes) predicts the correlation coefficient rate for the nodes (genes):(2)$$E\left(m,n\right)=\frac{N(G_{m}\cap G_{n})}{N\left(G_{m}\cap G_{n}\right)}$$

*G* indicates the total number of genes represented as nodes, and *E* denotes the number of connections among genes represented as edges. To cross-check illness comorbidity relationships, we used the R programs comoR [36] and POGO [37].

### 2.3. Enrichment Analysis for Significant Gene Ontology and Molecular Pathway Selection

Previously, gene expression profiling generally consisted of a group of genes corresponding to either healthy or affected samples, enlisted in a list *L* as per their differential expression. A meaningful understanding of this list was extracted. However, in a given biological process, it may provide an insufficient number or an excessive number of statistically significant genes that might fluctuate from one dissertation to another for a given batch of genes [38,39]. However, enrichment analysis denotes a normalized set of genes that employs previously identified molecular pathways or gene expression arrays. Moreover, it defined the group of genes associated with the different genotypes (phenotypes) hypothesis [40].

EnrichR was employed to acquire a deeper insight into the biological pathways and Gene Ontology (GO) terms associated with GBM in relation to I. stroke and mm [41]. It conducts GSEA to classify the DEGs’ corresponding pathways and GOs. Compared with a catalog of well-annotated gene sets, such as pathway analysis, it facilitates observing the functional relevance of the given gene set. The pathway is the molecular biology concept, which defines an artificial condensed process model within a cell or tissue [42]. A typical pathway model begins across an external signaling molecule by provoking a specific receptor that triggers a string of proteins connected with each other [43]. The Gene Ontology (GO) is a computational paradigm for representing gene (protein) functions as well as their related connections towards other genes [44]. The hierarchical arrangement of the GO makes it possible to compare proteins annotated with different meanings in ontology as well as have relationships with each other. We focused on four different pathway databases: KEGG [45], BioCarta [46], Reactome [47], and Wiki-Pathways [48]; and biological Process (BP) from Gene Ontology (GO) domain [49].

### 2.4. Analysis of Protein–Protein Interactions (PPIs)

The PPIs are central to all cellular/molecular mechanisms since they constitute the physical interactions between two or even more protein components [50]. We used data from the STRING database [51] and Network Analyst [52] to create PPI networks centered on the connections among various proteins. We used the String Interactome repository from “String-db.org” (accessed on 28 October 2014) with a confidence level of 800 and topological criteria such as degree >15 [51]. Proteins are denoted by colored circles/nodes; conversely, connections of the proteins are characterized by edges.

### 2.5. Analysis of Transcription Factors (TFs) and microRNAs (miRNAs)

We discovered DEGs-TFs, which regulate the identified significant genes (identified from transcriptomic differential analysis) not only at their correct period but also at their suitable volume in a cell throughout the cell’s/organism’s lifetime, and are responsible for determining the transformation of genetic information from DNA to mRNA at the transcription level. Furthermore, gene-miRNAs were also discovered in order to help researchers by giving insight into the regulatory biomolecules that determine and control RNA splicing and expression of genes at their post-transcriptional level.

EnrichR was deployed to identify the DEGs-TFs and microRNAs [41]. The DEGs-TFs relationship was identified and studied using the JASPAR database [53] and ENCODE [54,55], whereas miRNA-DEGs interactions are found using a well-known database called TarBase [56] and miRTarBase [57]. The topological investigation was carried out using Cytoscape’s Network Analyzer and Network Analyst [58,59].

### 2.6. Drug Prediction

Network Analyst was used to identify the possible medications for treating glioblastoma and its associated diseases. The drug was predicted using the DrugBank database version 5.0. [60]. A list of protein–drug interactions was made based on statistical importance. Two protein–drug interactions were predicted for two pairs of cases, such as glioblastoma and I. stroke, and glioblastoma and mm. In our study, we utilized highly interacted shared proteins (hub proteins) found from both pairs of cases.

### 2.7. Description of the Experimental Methodology

Figure 1 summarizes the network-based systemic and computational framework for evaluating differentially expressed human genes due to the association among diseases. The R code was used to introduce the optimized pipeline, and the implementation is accessible from our Github repository: https://github.com/hiddenntreasure/glioblastoma, accessed on 11 July 2022.

To identify hypothesized selective biomarkers between GBM and I. stroke and GBM and mm, we used gene expression analyses using limma. Moreover, we extracted signaling pathways and GO terminologies from various databases, as well as protein–protein interactions (PPIs), transcription factors (TFs) of genes, and gene-MicroRNAs (miRNAs) that are related to the derived biomarkers. Our network-based method was cross-checked with the three gold standard databases, namely, DisGeNET, db-GaP, and Rare-Diseases-AutoRIF, to validate our biomarker genes and pathways.

## 3. Result Analysis

### 3.1. Evaluation of Gene Expression

“Expression profiling by high-throughput sequencing” (or RNA-seq) data of GBM was reviewed from the NCBI to categorize and comprehend the gene enrichment that could influence the development of I. stroke and mm. However, due to the unavailability of moyamoya’s RNA-seq data, we collected “expression profiling by array” (or microarray) data of moyamoya.

A well-known project called Bioconductor established R packages called Limma and DESeq2 for microarray and RNA-seq data. We used it to perform expression profiling and found 3585 DEGs in glioblastoma with a *p*-value less than 0.05 and an absolute logFC greater than 1. Whereas 1038 genes are upregulated due to foreign signals increasing the cellular process factor in all genes, 2547 genes are downregulated due to the same component decreasing markedly. Following the statistical study, we identified the most significant DEGs for each disease, such as I. stroke and moyamoya. Table 1 illustrates that 1465 significant DEGs were found in I. stroke, whereas the expression increased (up-reg) in 1120 genes and expression decreased (down-reg) in 345 genes; similarly, 1382 significant DEGs were found in mm, whereas the expression increased (up-reg) in 715 genes and expression decreased (down-reg) in 667 genes. The GSE accession numbers for the selected study are GSE106804 [28], GSE56267 [31], and GSE131293 [32] for glioblastoma, ischemic stroke, and moyamoya, respectively, as shown in Table 1.

Due to the proper data availability, we took the dataset from three different cells: extracellular vesicle, cortical ischemic stroke tissue, and neural crest stem cell for GBM, I. stroke, and mm, respectively (Table 1, column 3). However, the findings still show insightful outcomes for our projected hypothesis. Column 2 in Table 1 demonstrates the RNA sequencing technology used to identify the transcriptomic data for each disease in our study. The number of samples for both cases and controls is an essential identifier in identifying associations among diseases because the increasing number of samples enhances the computational power of a dataset. In our study, moyamoya has only three samples, both for control and case, which is the least, whereas the other two diseases have at least six samples for either side. The overall up- and downregulated genes are quite balanced for moyamoya, though not for GBM and I. stroke.

### 3.2. Identified Enriched Pathways and Gene Ontology Terminologies

Pathway enrichment analysis was implemented to better understand the molecular mechanisms/processes that underlie all complicated diseases. Using EnrichR [41], a bioinformatics resource, we conducted a comparison-based enrichment analysis to classify overexpressed pathways in our relationship (GBM and I. stroke; or GBM and MM), and the analysis was performed on top of three different databases (Wikipathways (human-2019) [61], BioCarta (2016) [41], and KEGG (human-2019) [62]) in our experiment. The pathway enrichment experiments were performed using the common DEGs between GBM and its associated diseases (I. stroke and mm). We carried out regulatory research to learn more about the molecular mechanisms that play a role in this comorbidity. Our research identified overexpressed pathways in which DEGs are identified in various disorders and categorized them based on their functional importance. Manual curation was used to limit pathways considered greatly enriched in the typical DEG sets with *p*-value criteria. The criteria denote that the *p*-value must be less than 0.05. EnrichR discovered major pathways from KEGG, WikiPathways, and BioCarta databases that are significantly linked to DEGs that are common between GBM and I. stroke pair and GBM and mm pairs. Using the shared 50 genes between GBM and mm, we obtained 149 shared pathways, among which 20 are significant, considering the *p*-value (<0.05). Similarly, 59 genes are common between GBM and I. stroke; we obtained 217 signaling pathways common between them, and 68 are highly expressed (significant pathways). Thus, ascending sorting of *p*-value implied retrieving the top 15 significant pathways between (a) GBM and I. stroke—Table 2 and (b) GBM and mm—Table 3.

We also discovered highly expressed Gene Ontology (GO) terms, especially for identifying molecular events associated with a disease. Therefore, popular DEGs between two diseases were employed to obtain the list of GOs associated with a disease. The Enrichr was used to find GO terms enriched by shared DEGs. Enrichr introduces biological processes (BP-2016) that are linked to DEGs so that they can be grouped into functional categories [63,64]. Hence, this helps us learn more about the molecular processes and biological relevance of DEGs. It was then narrowed down to only those processes and terms with a relative *p*-value below 0.05. Between GBM and mm, 503 GO terminologies are shared, where 138 are significant GO terms (*p*-value < 0.05). Likewise, GBM and I. stroke have 652 shared GO terms, among which 193 are significant. Table 4 and Table 5 summarize the biological processes discovered, representing only the top 15 GO terms of BP-2016 for both pairs (a) GBM and I. stroke and (b) GBM and mm.

### 3.3. Protein–Protein Interactions (PPIs) Analysis

With the use of online-based tools such as STRING and Network Analyst, we built putative PPI networks utilizing our enriched common disease genes. PPIs try to compensate for the organism’s so-called interactomics, in which abnormal PPIs cause numerous illnesses. One or more typically linked protein subnetworks are reported to be represented by two diseases. PPI analysis revealed strongly interacting proteins employing topological criteria, such as a degree higher than 15°. Figure 2A shows the PPI network between GBM and mm. The network includes 59 nodes (genes) and 29 edges; the PPI network’s enriched *p*-value is 0.232. Figure 2B demonstrates the PPI network for GBM and I. stroke, where there are 55 nodes and 65 edges, where the PPI-network’s enriched *p*-value is 1.11 × 10^−16^.

The cytoHubba module was used to explore the most significant hub-proteins based on the simplified PPI networks developed previously [65]. We found 14 hub proteins between GBM and mm using four cytoHubba algorithms, and they are MCC, DMNC, Degree, and EPC (as shown in Figure 3, 11 hub proteins are shared by all the algorithms: *CASP1*, *PSMA3*, *PSMA4*, *TNPO1*, *PSMA2*, *MEFV*, *PSMA6*, *PSMB9*, *PSMB1*, *PYCARD*, and *YME1L1*, and three are shared by Degree and MCC: *AK7*, *POLR3B*, and *POLR3E*.

Similarly, we found 26 hub proteins between GBM and I. stroke, as shown in Figure 4. All the four cytoHubba algorithms share 21 hub proteins: *COL1A1*, *ANXA2*, *PPBP*, *SPARC*, *TIMP1*, *SERPINE1*, *PECAM1*, *HLA-DRA*, *CXCR4*, *ALOX5AP*, *S100A12*, *BCL2A1*, *HLA-DQA1*, *LCP2*, *GNB5*, *S100A8*, *PLEK*, *ARHGEF9*, *LCP1*, *IL2RG*, and *SLA*; two are shared by Degree, MCC, and EPC: *TREML1* and *F11R*; two are shared by Degree, EPC, and DMNC: *SERPINA1* and *NCF2*; and only one is shared by Degree and EPC: *ANKRD1*. Although further research into the activities of these newly discovered hub proteins is needed, they might be potential therapeutic targets.

### 3.4. Determination of the DEGs’ Transcriptional and Post-Translational Regulators

Transcription factors (TFs) are nothing but proteins that govern the expression of the identified significant genes in our case. In other words, the transcriptional process converts genes into RNA or protein products. Transcription factors are found in all living organisms and regulate gene expression. TF genes are significant because they regulate a variety of biological processes [66,67]. miRNA plays a vital role in cellular processes and biochemical and molecular functions [68]. As a result, changes in miRNA levels (enriched miRNA) may affect metabolic processes, signal transmission, and transcription [69]. According to this study, microRNAs play a role in various diverse biological characteristics related to glioblastoma, such as cell growth, incursion, glioma stem cell activity, and angiogenesis (blood vessel formation) [70]. Additionally, miRNA functions may aid in elucidating the dysregulated signaling pathways and provide insight into the development of novel therapeutic and diagnostic procedures [71].

In Figure 5, we visualize the DEGs-TFs and DEGs-miRNAs that controlled the gene expression and multiple BP in a patient with glioblastoma and/or moyamoya. The transcription- and post-transcription-level regulatory genes between GBM and mm include *PCCA*, *YME1L1*, *PEX26*, *XYLT2*, *ZNF71*, *POLR3E*, *ZNF76*, *NRF1*, *TFDP1*, *ZNF610*, *ZNF101*, *TNPO1*, *CNOT7*, *TMCO3*, *TGIF2*, *CEP57L1*, *IRF1*, *MAPK13*, *CREB3L1*, *SOX13*, *LIMD1*, *RNF8*, *PSMB9*, and *FAM111A*. The miRNAs, non-coding gene products, similar for GBM and mm include: miR-522-5p, miR-1-3p, miR-146a-5p, miR-6499-3p, miR-34a-5p, miR-7977, miR-6778-3p, miR-107, miR-374a-5p, miR-16-5p, miR-27a-3p, miR-124-3p, miR-128-3p, miR-155-5p, miR-92a-3p, miR-24-3p, and miR-455-3p. Figure 6 represents the TF-gene and gene-miRNA that regulate mechanisms of gilobastoma and I. stroke. TF-genes that are included between GBM and I. stroke are *PPARG*, *TGIF2*, *NFIC*, *SRF*, *IRF2*, *GATA3*, *GABPA*, *GATA4*, *NCF2*, *MT2A*, *HOXA5*, *ALOX5AP*, *KDM1A*, *MLKL*, *SSRP1*, *S100A8*, *NFKB1*, *GABRA1*, *CXCR4*, *TMEM71*, *YY1*, *RCOR2*, *HS3ST3B1*, *BAZ1A*, *F11R*, *PCSK5*, *E2F1*, *SERPINA1*, *SREBF2*, *PRRC2A*, *LIMS1*, *GATA2*, *FOXA1*, *ARHGEF9*, *TP53*, *CREB1*, *ZEB1*, *GNB5*, *FOXC1*, *SMAD5*, *PSMB9*, *ARL17A*, *IRF1*, *PNP*, *MLX*, *ANXA2*, *HMG20B*, *SERPINE1*, *FOXL1*, *TFDP1*, *MTHFD2*, *COL1A1*, *HDGF*, *ZNF76*, *ATF1*, and *CREB3L1*. The miRNAs are miR-27a-3p, miR-6817-3p, miR-124-3p, miR-1-3p, miR-16-5p, miR-129-2-3p, miR-129-5p, miR-26b-5p, miR-122-5p, miR-355-5p, miR-6778-3p, and miR-192-5p.

### 3.5. Analysis of the Predicted Drugs

We predicted drugs using shared proteins that resulted from our analysis. A web tool called Network Analyst was employed, which collected data from the DrugBank 5.0 database. We utilized 26 hub proteins shared between GBM and I. stroke to discover the drugs as represented in Figure 7B. The protein–drug interaction (Figure 7A) has ten nodes, including two genes (*ANXA2* and *SERPINE1*) and eight chemical compounds (Alteplase, Tenecteplase, Urokinase, Plasmin, Troglitazone, Drotrecogin alfa, Anistreplase, and Reteplase). Similarly, we used 14 shared hub proteins from GBM and mm for drug prediction, as shown in Figure 7A. It involves nine nodes, including CASP1 gene and eight chemical compounds (VX-765, IDN-6556, LAX-101, Pralnacasan, Minocycline, 3-[6-[(8-HYDROXY-QUINOLINE-2-CARBONYL)-AMINO]-2-THIOPHEN-2-YL-HEXANOYLAMINO]-4-OXO-BUTYRI ACID, 3-[2-(2-BENZYLOXYCARBONYLAMINO-3-METHYL-BUTYRYLAMINO)-PROPIONYLAMINO]-4-OXO-PENTANOIC ACID, and 1-METHYL-3-TRIFLUOROMETHYL-1H-THIENO[2,3-C]PYRAZOLE-5-CARBOXYLIC ACID (2-MERCAPTO-ETHYL)-AMIDE).

### 3.6. Validation of Transcriptomic Potential DEGs

We validated our potential DEGs of transcriptomic analysis by using literature-based disease-gene association datasets such as DisGeNET [72], dbGaP [73], and Rare-Diseases-AutoRIF. The data were created and validated using the previous study, including the biomarker genes corresponding to diseases. In order to assess the shared genes’ statistical significance and validate our findings, we employed EnrichR [41], an online program. EnrichR utilized the shared genes with the disease-associated gene database to discover the relevant data. Even though EnrichR gives disease-gene information for a variety of disorders, we only take into account the information pertaining to the diseases we identified.

We further confirmed our findings by reviewing previous publications that discovered biomarkers for the diseases. The literature associated with each gene is included in Table 6 and Table 7. Ultimately, we created a diseasome network of GBM and its associated neurological and vascular disorders, as shown in Figure 8. We developed this association map from the gold benchmark database and previous literature review using Cytoscape [58].

## 4. Discussion

According to current research, it is clear that glioblastoma is the most aggressive type of brain cancer. It is also known that glioblastoma is responsible for an increased risk of developing ischemic stroke [9]. Similarly, moyamoya disease develops in brain tumor patients due to cranial irradiation during radiation therapy [128]. Thus, it is possible that ischemic stroke and moyamoya may be formed in glioblastoma patients.

Hence, our study aims to identify genetic relationships between glioblastoma and ischemic stroke as well as glioblastoma and moyamoya. Thus, doctors should be concerned about ischemic stroke and moyamoya in glioblastoma patients. The bioinformatics approach may comprehensively understand the molecular mechanisms in the specified disease progression. In this study, we carried out an investigation on transcriptomic profiles of ischemic stroke, moyamoya, and glioblastoma (as shown Figure 1). Moreover, we predicted the therapeutic drugs for the associations.

To determine if any significant dysregulation existed, we performed differential expression analysis (DEA) followed by identifying shared genes for glioblastoma, moyamoya, and glioblastoma, ischemic stroke (as shown in Figure 9 and Figure 10, respectively). We also demonstrated diseasome network (in Figure 8), pathways (represented in Table 2 and Table 3), Gene Ontology (GO) (as shown in Table 4 and Table 5), protein–protein interactions (in Figure 2), hub–protein interactions (in Figure 3 and Figure 4, respectively), drug–protein interactions (shown in Figure 7), and transcription factor gene interactions and gene miRNA interactions (represented in Figure 5 and Figure 6 separately). In addition, a transcriptomic dataset (RNA-Seq) was collected from ischemic stroke, moyamoya, and glioblastoma patients and healthy individuals (as shown in Table 1). We also verified our candidate genes by previous literature published in various journals (as shown in Table 6 and Table 7, respectively). The flow diagram of our methodology is visually represented and outlined with proper direction in Figure 1.

At first, we focused on eight hub genes, named CXCR4, ANXA2, SPARC, SERPINA1, NCF2, COL1A1, LCP2, and IL2RG, that are highly expressed in glioblastoma and ischemic stroke (as shown in Figure 4). Astrocytes, neurons, bone marrow-derived cells, neural progenitor cells, and microglia1 all have CXCR4, and CXCR4 expression is regulated in a variety of clinical situations, including brain I. stroke [129]. There are many other elements of brain tumor biology where CXCR4 is responsible for developing glioblastoma, including cancer-related cells’ ability to resist radiotherapy and chemotherapy, and there are migration and production of the blood supply to the tumor [130]. As a potential candidate for invasion-boosting, and enriched in its initial stage of developing a brain tumor, SPARC has been found and described [84]. SERPINA1 was shown to be expressed in glioma tissue samples [131]. The latest study demonstrated six-fold enrichment of SERPINA1 in human atherosclerotic in contrast to healthy ones to verify the involvement of SERPINA1 in atherosclerosis [132]. SERPINE1 has been discovered as a regulator of GBM cell dispersion. Prevention of GBM tumor growth and invasiveness in the brain was achieved by knocking down the SERPINE1 [112]. Both the CGA cancer database and clinical evidence reveal that relatively high enrichment of NCF2 genes is associated with a bad outcome in glioblastoma patients [119]. Overexpression or knockdown of COL1A1 was used to examine the effect on glioma cell proliferation of COL1A1 [133]. LCP2 and IL2RG are not reported. In future research, these genes can be further studied to prevent ischemic stroke in GBM patients. Similarly, a hub gene named CASP1 was found between GBM and moyamoya, as shown in Figure 3. CASP1 plays an important role in upregulating the development of glioma [134,135]. Researchers can work on this gene to avoid developing moyamoya disease in GBM patients.

For pathways, there are two pathways, named antigen processing and presentation and serotonin and anxiety, that are common in glioblastoma, ischemic stroke, and moyamoya. In IDH-wildtype gliomas, the antigen processing and presentation (APP) score is linked with the immunological score [136]. Antigen processing and presentation, DC pathway, cytokine pathway, and IL-12 pathway were increased in the intracranial arteries of patients with mm in this study [137]. The serotonin and anxiety pathway is known as the monoaminergic system [138]. In addition, ischemic brain injury alters this route, and the monoaminergic system may be a potential therapeutic target for stroke [139]. Therefore, these four pathways can be a therapeutic target in order to prevent ischemic stroke and moyamoya associated with glioblastoma patients. In addition, a pathway named leukocyte transendothelial migration is activated, which is validated by a previous study between glioblastoma and ischemic stroke. It is possible that an aberrant immunological condition and the development of GBM are associated, and the leukocyte transendothelial migratory pathway might be an indicator of that [140].

According to the information presented above, our technique has the potential to disclose some of the essential mechanisms that underlie disease, as well as generate unique theories about disease mechanisms and identify new biomarkers for disease. Genetic data analysis is expected to be crucial for improving predictive medicine and uncovering pathways connecting with glioblastoma, ischemic stroke, and moyamoya, as well as identifying potential therapeutic targets.

We made an effort to use prior research to validate each of our findings. However, there is still a need for more in vivo and in vitro research. Due to their complexity, the doctor must be concerned about ischemic stroke and moyamoya in glioblastoma patients. Moreover, the prevention of ischemic stroke and moyamoya can be made possible by inactivating mentioned pathways using the predicted drug.

A few limitations open the way for further research, such as the availability of brain-related data from living organisms. Moreover, more specific clinical- and gene-level research is required to better understand the complications by analyzing the candidate biomarkers found in this work.

## 5. Conclusions

The current study used a statistical technique on the transcriptomic data to uncover the shared significant genes that are highly enriched among glioblastoma, ischemic stroke, and moyamoya patients. The study of the significant gene sets revealed the associated dysregulated pathways that were also highly enriched. Protein–protein interactions, regulatory TFs from the survey of TF–gene interactions, and miRNAs from gene–miRNA interactions were obtained by comparing the overlapped DEGs with distinct biomolecular interaction networks and databases. Most of the transcription factors and microRNAs discovered in this study are novel; no prior studies have implicated these genes or pathways in developing these disorders or their connections. More studies still need to be performed to validate these molecular signature biomarkers. This study looked at candidate genes at protein and RNA levels, such as TFs, mRNAs and miRNAs, the pathway, and the GO terminologies. Finally, we predicted the potential drugs for the associations. Moreover, the results were validated using gold benchmark databases and published literature. These results show that genes in glioblastoma are more or less active in people with ischemic stroke and moyamoya, which could help explain these diseases. It also demonstrates how to find functional relationships between ischemic stroke and moyamoya, explaining why they are linked to glioblastoma.

## Figures and Tables

**Figure 1 pharmaceutics-14-01573-f001:**
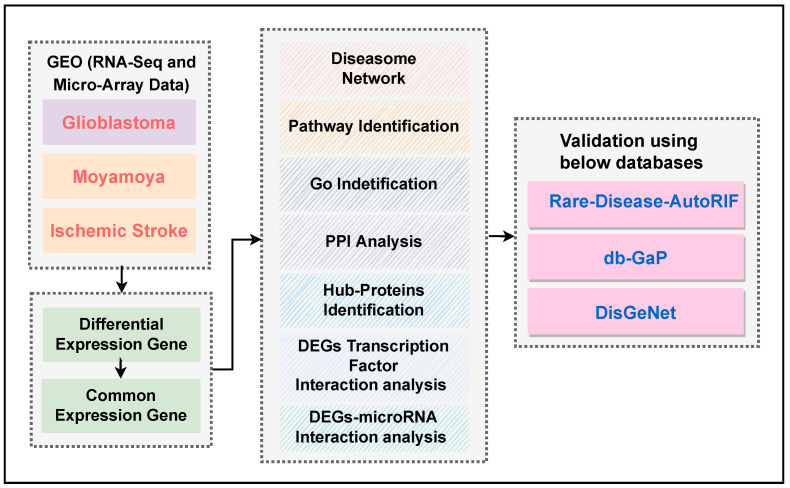
Demonstration of the work flow of our hypothesized methodology.

**Figure 2 pharmaceutics-14-01573-f002:**
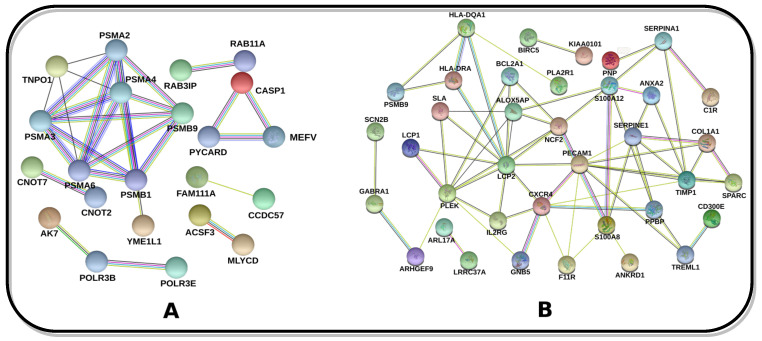
Protein–protein interactions found using the shared significant genes. (**A**) PPI between Glioblastoma and moyamoya. (**B**) PPI between glioblanstoma and Ischemic stroke.

**Figure 3 pharmaceutics-14-01573-f003:**
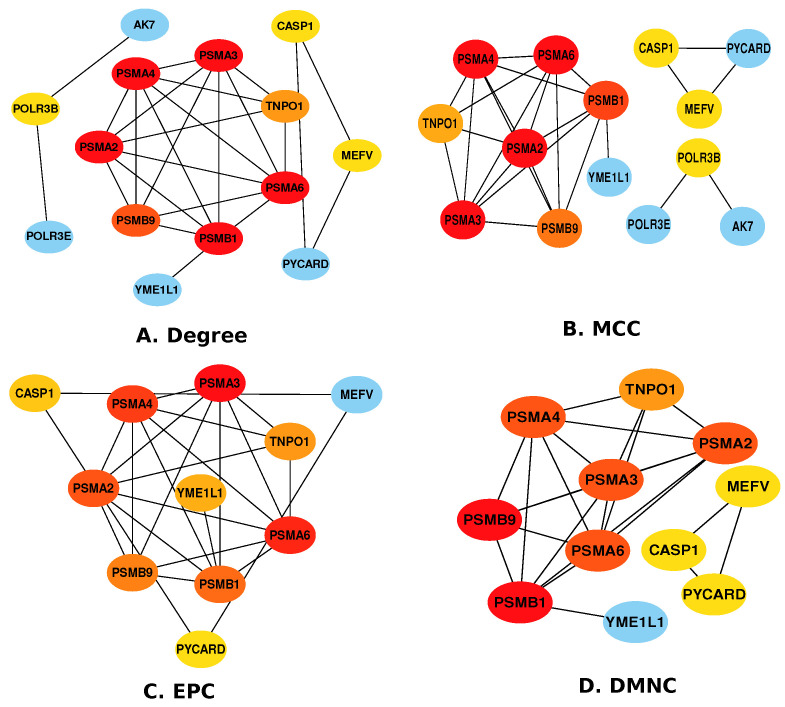
Hub proteins identified using four different cytoHubba algorithms between Glioblastoma and Moyamoya.

**Figure 4 pharmaceutics-14-01573-f004:**
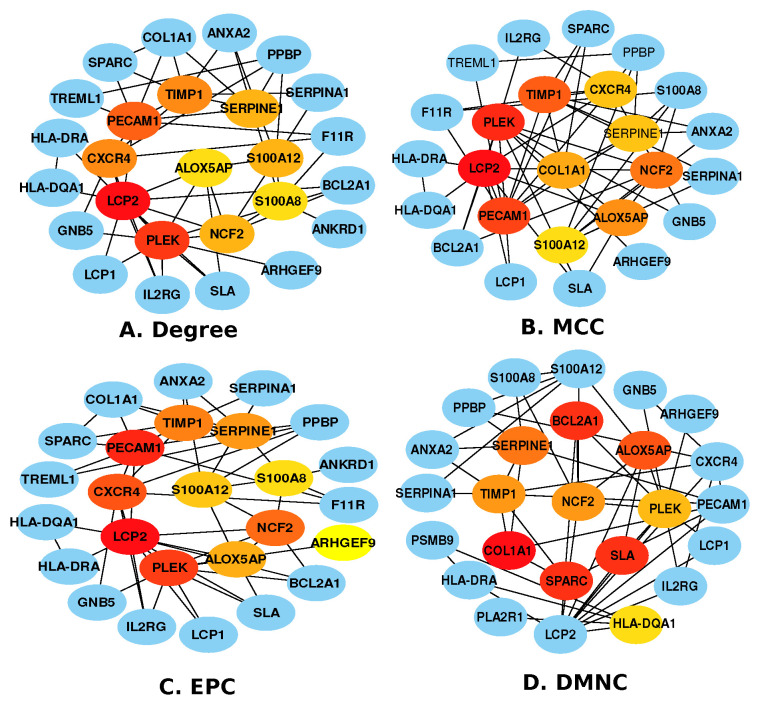
Hub proteins identified using four different cytoHubba algorithms between glioblastoma and Ischemic stroke.

**Figure 5 pharmaceutics-14-01573-f005:**
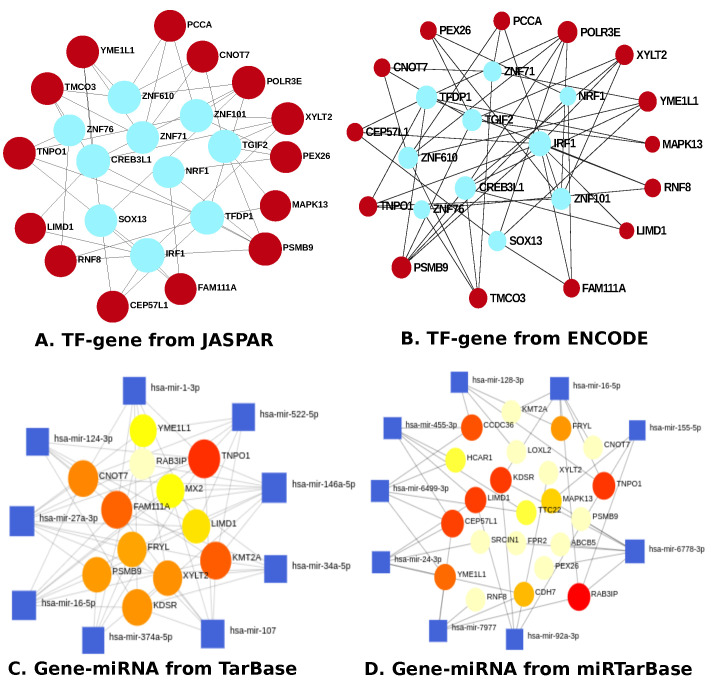
Visualization of the DEGs-TFs and miRNAs interactions between glioblastoma and moyamoya using various databases: JASPER and ENCODE for TF-gene; TarBase and miRTarBase for gene-miRNA.

**Figure 6 pharmaceutics-14-01573-f006:**
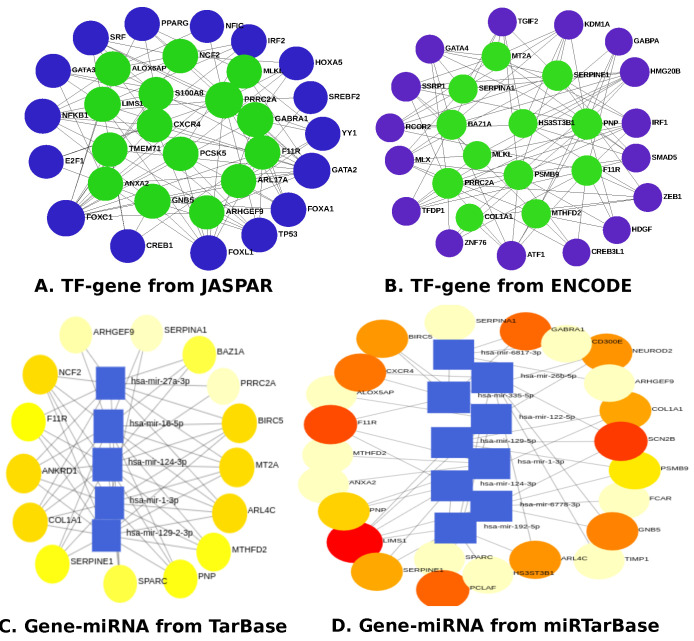
Representation of the DEGs-TFs and miRNAs interactions between glioblastoma and I. stroke using various databases: JASPER and ENCODE for TF-gene; TarBase and miRTarBase for gene-miRNA.

**Figure 7 pharmaceutics-14-01573-f007:**
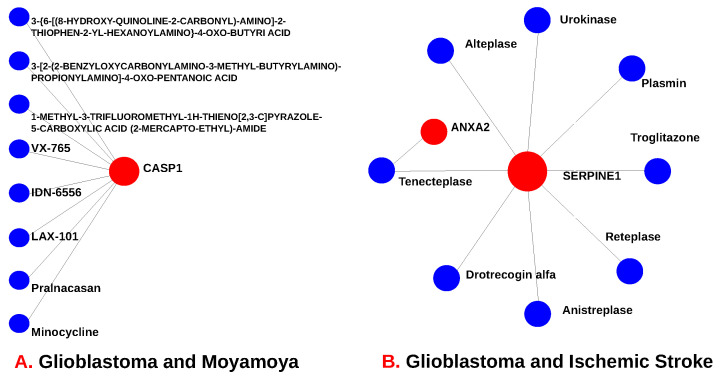
This figure shows the drug–protein interaction. (**A**) Glioblastoma and moyamoya. (**B**) Glioblastoma and ischemic stroke.

**Figure 8 pharmaceutics-14-01573-f008:**
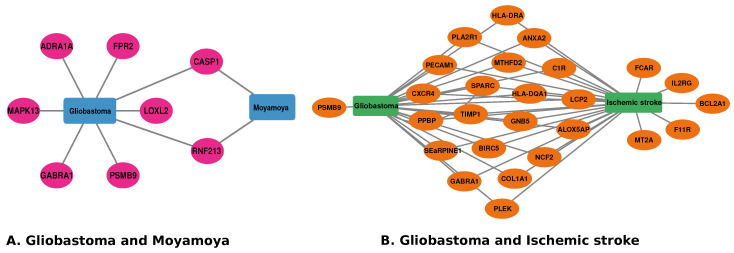
Diseasome network for our study, where rectangle nodes define the diseases and ellipses nodes define the genes associated with corresponding disease. (**A**) Diseasome network between GBM and MM. (**B**) Diseasome network between GBM and I. stroke.

**Figure 9 pharmaceutics-14-01573-f009:**
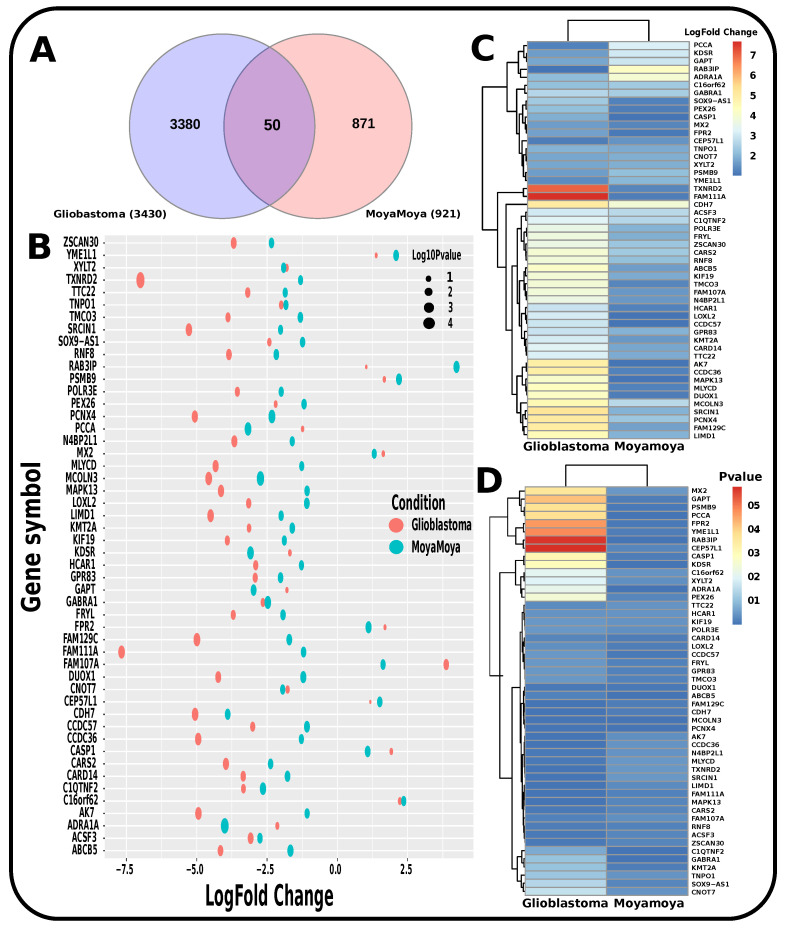
Representation of the significant genes found to be common for glioblastoma and moyamoya by transcriptomic-based investigation. (**A**) Venn diagram shows the significant common biomarker genes. (**B**) Log-fold changes and *p*-value combined to generate a bubble plot for the common significant genes. (**C**) Heatmap that demonstrates the LogFC. (**D**) Heatmap that demonstrates the *p*-value.

**Figure 10 pharmaceutics-14-01573-f010:**
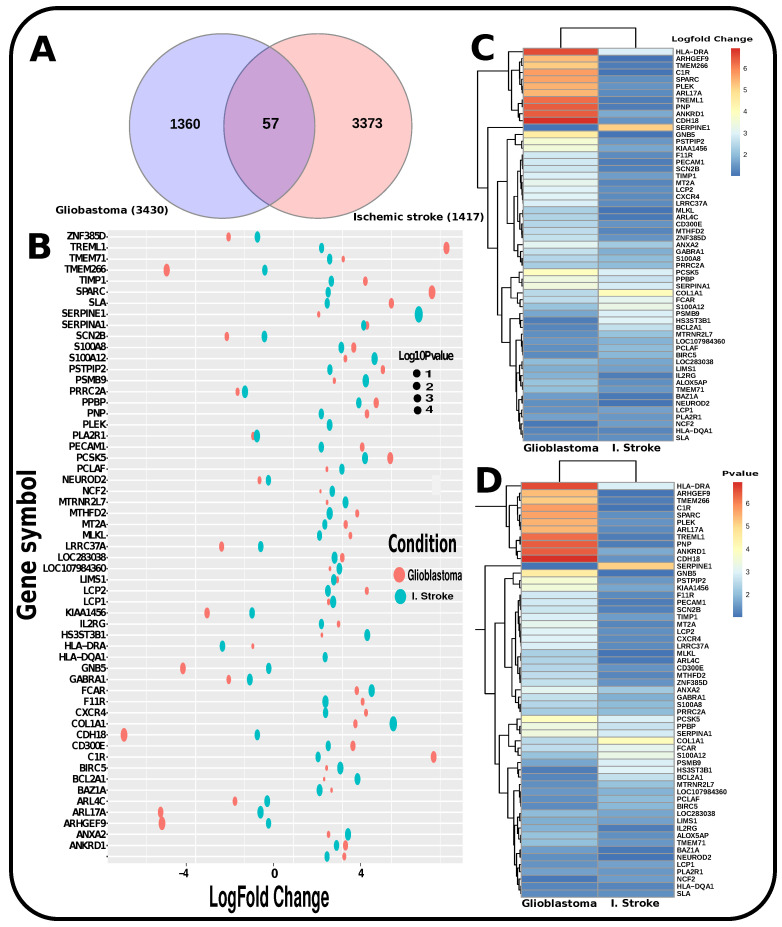
Representation of the significant genes found to be common for glioblastoma and I. stroke by transcriptomic-based investigation. (**A**) Venn diagram shows the significant common biomarker genes. (**B**) Log-fold changes and *p*-value combined to generate a bubble plot for the common significant genes. (**C**) Heatmap that demonstrates the LogFC. (**D**) Heatmap that demonstrates the *p*-value.

**Table 1 pharmaceutics-14-01573-t001:** Detailed information about the selected transcriptomic datasets from NCBI that meet all the criteria.

Disorder	Source	Dataset	Raw	Case	Control	Significant Up	Significant Down
Name	Tissues/Cells	Accession No.	Genes	Samples	Samples	Reg. Genes	Reg. Genes
Glioblastoma	Extracellular	GSE-106804	59,171	13	6	1038	2547
	Vesicle						
Ischemic Stroke	Cortical ischemic stroke tissue	GSE-56267	28,089	7	6	1120	345
Moyamoya	Neural crest	GSE-131293	54,675	3	3	715	667
	stem cell						

**Table 2 pharmaceutics-14-01573-t002:** List of top-15 highly-expressed pathways between GBM and I. stroke.

Pathway Name	*p*-Value	Database
Phagosome	4.6 × 10^−6^	KEGG-orthologs
Staphylococcus aureus infection	4.21 × 10^−5^	KEGG-orthologs
Photodynamic therapy induced HIF-1 survival signaling	0.000159	Wiki-Pathways
Leukocyte transendothelial migration	0.000292	KEGG-orthologs
Intestinal immune network for IgA production	0.000346	KEGG-orthologs
Antigen Processing and Presentation	0.0005171	BioCarta
Complement and Coagulation Cascades WP558	0.000605	Wiki-Pathways
Lung fibrosis WP3624	0.00077	Wiki-Pathways
Cell adhesion molecules (CAMs)	0.000777	KEGG-orthologs
IL 4 signaling pathway	0.000818	BioCarta
Inflammatory bowel disease (IBD)	0.000845	KEGG-orthologs
miR-509-3p alteration of YAP1/ECM axis	0.001055	Wiki-Pathways
Serotonin and anxiety WP3947	0.00105	Wiki-Pathways
Leishmaniasis	0.001232	KEGG
Th1 and Th2 cell differentiation	0.00230	KEGG-orthologs

**Table 3 pharmaceutics-14-01573-t003:** List of top-15 highly-expressed pathways between GBM and moyamoya.

Pathway Name	*p*-Value	Database
Serotonin and anxiety	0.000813	Wiki-Pathways
Propanoate metabolism	0.002895921	KEGG-orthologs
GPCRs, Class A Rhodopsin-like WP455	0.003858665	Wiki-Pathways
Leucine, valine, and isoleucine degradation	0.00642032	KEGG-orthologs
Amyotrophic lateral sclerosis	0.007222503	KEGG-orthologs
Neuroactive ligand-receptor interaction	0.010018848	KEGG-orthologs
Cytosolic DNA sensing pathway	0.010854501	KEGG-orthologs
D4-GDI Signaling Pathway	0.014908285	BioCarta
Pertussis	0.015517163	KEGG-orthologs
Peroxisome	0.018324143	KEGG-orthologs
Salmonella infection	0.019588053	KEGG-orthologs
Cardiac Protection Against ROS	0.027165345	BioCarta
C-type lectin receptor signaling pathway	0.027901009	KEGG-orthologs
Antigen Processing and Presentation	0.029598762	BioCarta
AMPK signaling pathway	0.0362706	KEGG-orthologs

**Table 4 pharmaceutics-14-01573-t004:** List of significant GO terminologies that are common between GBM and I. stroke.

Biological Process	*p*-Value	GO Id
Platelet degranulation	0.0000000607	GO:0002576
Regulated exocytosis	0.000000204	GO:0045055
Cytokine-mediated signaling pathway	0.00000144	GO:0019221
Extracellular matrix organization	0.00000383	GO:0030198
Regulation of endopeptidase activity	0.0000421	GO:0052548
Neutrophil degranulation	0.0000602	GO:0043312
Neutrophil activation involved in immune response	0.0000638	GO:0002283
Neutrophil mediated immunity	0.0000676	GO:0002446
Replicative senescence	0.000432	GO:0090399
Neutrophil migration	0.000606	GO:1990266
Positive regulation of DNA damage response, signal transduction by p53 class mediator	0.00071	GO:0043517
Negative regulation of peptidase activity	0.000737	GO:0010466
Interferon-gamma-mediated signaling pathway	0.001049284	GO:0060333
Positive regulation of signal transduction by p53 class mediator	0.00105588	GO:1901798
Defense response to fungus	0.00105588	GO:0050832

**Table 5 pharmaceutics-14-01573-t005:** List of significant GO terminologies that are common between GBM and moyamoya.

Biological Process	*p*-Value	GO Id
B cell activation involved in immune response	0.000469	GO:0002312
Post-transcriptional gene silencing by RNA	0.000913	GO:0035194
Gene silencing by miRNA	0.002719256	GO:0035195
Fatty acid biosynthetic process	0.006162594	GO:0006633
Cell morphogenesis	0.009570011	GO:0000902
Regulation of viral genome replication	0.010854501	GO:0045069
Monocarboxylic acid biosynthetic process	0.012560892	GO:0072330
Lipid biosynthetic process	0.014004804	GO:0008610
Positive regulation of action potential	0.014908285	GO:0045760
Positive regulation of cardiac muscle contraction	0.014908285	GO:0060452
Astrocyte activation	0.014908285	GO:0048143
Negative regulation of type I interferon-mediated signaling pathway	0.014908285	GO:0060339
Acetyl-CoA biosynthetic process	0.014908285	GO:0006085
Regulation of hematopoietic stem cell differentiation	0.015517163	GO:1902036
Regulation of hematopoietic progenitor cell differentiation	0.015905777	GO:1901532

**Table 6 pharmaceutics-14-01573-t006:** Transcriptomic analysis identifies potential target genes in GBM and mm that have been verified by previous research.

Gene	Gliobastoma	Moyamoya
CASP1	Chen et al., [74]—2022	Kang et al., [75]—2010
GABRA1	D’Urso et al., [76]—2012	-
MLYCD	Avsar [77]—2021	-
CARD14	-	Constantin et al., [78]—2010
RNF213	Bao et al., [79]—2014	Fujimura et al., [80]—2014
LOXL2	Zhang et al., [81]—2020	-
HCAR1	Longhitano et al., [82]—2021	-
FPR2	Yang et al., [83]—2020	-

**Table 7 pharmaceutics-14-01573-t007:** Transcriptomic analysis identifies potential target genes in GBM and I. stroke that have been verified by previous research.

Gene	Gliobastoma	I. Stroke
SPARC	Golembieski et al., 1999 [84]	Baumann et al., 2009 [85]
C1R	Ma et al., 2021 [86]	Mitaki et al., 2021 [87]
PPBP	Lei et al., 2021 [88]	Katnik et al., 2016 [89]
PECAM1	Warrier et al., 2021 [90]	Beom et al., 2015 [91]
TIMP1	Aaberg-Jessen et al., 2009 [92]	Worthmann et al., 2010 [93]
COL1A1	Sun et al., 2018 [94]	Choi et al., 2019 [95]
FCAR	Hassan et al., 2017 [96]	-
MT2A	Sun et al., 2018 [94]	-
MTHFD2	Han et al., 2019 [97]	Kasiman 2012 [98]
LCP2	Li et al., 2016 [99]	Li et al., 2021 [100]
ALOX5AP	Liu et al., 2020 [101]	Bie et al., 2021 [102]
F11R	Hattermann et al., 2014 [103]	-
CXCR4	Cornelison et al., 2018 [104]	Bang et al., 2012 [105]
ANXA2	Tu et al., 2019 [106]	Li et al., 2021 [107]
IL2RG	Ogawa et al., 2018 [108]	-
PSMB9	-	Chen et al., 2021 [109]
PLEK	Hoelzinger et al., 2005 [110]	Zeng et al., 2015 [111]
SERPINE1	Seker et al., 2019 [112]	Bruno et al., 2021 [113]
BIRC5	Kim et al., 2016 [114]	Chon et al., 2016 [115]
HLA-DQA1	Urup et al., 2016 [116]	Zou et al., 2002 [117]
BCL2A1	-	Lin et al., 2021 [118]
NCF2	Wang et al., 2020 [119]	Zhou et al., 2021 [120]
GNB5	Xie et al., 2018 [121]	Jung et al., 2018 [122]
GABRA1	D’Urso et al., 2012 [76]	Feng et al., 2021 [123]
PLA2R1	Maruyama et al., 2021 [124]	Berchtold et al., 2021 [125]
HLA-DRA	Basta et al., 1998 [126]	Liu et al., 2021 [127]

## Data Availability

Links of our dataset are included in this manuscript.

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
