# Peer review of "Bioinformatics Strategies to Identify Shared Molecular Biomarkers That Link Ischemic Stroke and Moyamoya Disease with Glioblastoma"

_pharmaceutics, 2022, doi:10.3390/pharmaceutics14081573_

Round 1

Reviewer 1 Report

The authors investigated common molecular biomarkers between ischemic stroke or moyamoya disease and glioblastoma using bioinformatics techniques. The paper is interesting but a revision is needed.   The paper was not submitted using the template of the journal.   The references are not written in the American Chemical Society style required by the journal.   The paper would benefit from a moderate polishing of the English language. For example, the word ischemic is misspelled in the title.   In addition, the assistance of a neurologist/neurosurgeon/oncologist could be needed to clarify the background of the paper and check whether the medical information regarding the three disorders is correct. For example, in the abstract/introduction you talk about hemorrhagic stroke but then you talk about ischemic stroke. Please ask a colleague to check this part of the paper and you can acknowledge his help in the dedicated section.   Some of the information can also be presented as a supplementary file, the paper is very lengthy which makes it difficult to read.  

Author Response

The authors investigated common molecular biomarkers between ischemic stroke or moyamoya disease and glioblastoma using bioinformatics techniques. The paper is interesting but a revision is needed.  

  1. The paper was not submitted using the template of the journal. The references are not written in the American Chemical Society style required by the journal.

>Answer: Thank you for your observation. Hope the editorial section will make the changes if get accepted.

  1. The paper would benefit from a moderate polishing of the English language. For example, the word ischemic is misspelled in the title.

>Answer: Thank you for your observation. We have modified our manuscript and highlighted accordingly.

  1. In addition, the assistance of a neurologist/neurosurgeon/oncologist could be needed to clarify the background of the paper and check whether the medical information regarding the three disorders is correct. For example, in the abstract/introduction you talk about hemorrhagic stroke but then you talk about ischemic stroke. Please ask a colleague to check this part of the paper and you can acknowledge his help in the dedicated section.

>Answer: Thank you for your observation. We have gone through the introduction and we think its okay considering background of research. Hope you will like it.

  1. Some of the information can also be presented as a supplementary file, the paper is very lengthy which makes it difficult to read.

>Answer: Thank you for your observation. We have included only the most significant filtered results in our manuscript and highlighted accordingly in our manuscript.

Reviewer 2 Report

In this work the authors utilize publicly available datasets to study pathogenesis of glioblastoma, ischemic stroke and Moyamoya disease. They use a series of well-established algorithms to identify differentially expressed genes, deregulated gene networks, protein-protein interactions, analysis of transcription factors and microRNAs.

The authors perform a series of binary comparisons between two disease states each time, identifying common affected genes and pathways, protein interactions and transcription factors and miRNAs.

In the present form, I believe that the results are rather preliminary and unsuitable for publication. I think that the authors should provide a more integrative view, trying to identify common pathogenetic mechanisms in all three diseases. Moreover, they should also try to validate at least part of) their findings in a relevant biological system (e.g. by KO or overexpressing some of the DG genes in a cellular model). 

Moreover they should:

1) consider using more datasets with common features: the dataset chosen for glioblastoma is from extracellular vesicles that can differ significantly from actual malignant cells. Also the authors should avoid comparing data from whole transcriptome analysis with data from micro-arrays.

2) explain why the number of genes in Raw Data from the glioblastoma dataset is so low compared to the other two diseases (Table 1). It is not clear in the text which data was retrieved from micro-arrays: Glioblastoma or moyamoya.  

3) provide details on how the analysis were performed (which versions of the algorithms and what settings were selected) and also upload the code generated in github

4) evaluate whether some of the emerging targets are druggable

5) edit the text, with the aim to produce a more compact manuscript focusing more on the results obtained and their biological and clinical relevance.

6) revise several spelling, grammar and syntax errors (some of them even in the manuscript title)

Author Response

In this work the authors utilize publicly available datasets to study pathogenesis of glioblastoma, ischemic stroke and Moyamoya disease. They use a series of well-established algorithms to identify differentially expressed genes, deregulated gene networks, protein-protein interactions, analysis of transcription factors and microRNAs.

The authors perform a series of binary comparisons between two disease states each time, identifying common affected genes and pathways, protein interactions and transcription factors and miRNAs.

In the present form, I believe that the results are rather preliminary and unsuitable for publication. I think that the authors should provide a more integrative view, trying to identify common pathogenetic mechanisms in all three diseases. Moreover, they should also try to validate at least part of) their findings in a relevant biological system (e.g. by KO or overexpressing some of the DG genes in a cellular model).

Moreover they should:

  1. consider using more datasets with common features: the dataset chosen for glioblastoma is from extracellular vesicles that can differ significantly from actual malignant cells.

>Answer: Thank you  for  your observation. We have made the changes and highlighted accordingly in our manuscript.  We have  considered few criteria to select a dataset. They are:

  1. The dataset should be obtained from a human species.
  2. There should be control samples (healthy) and case samples (patient's).
  3. We have discarded repeated datasets, unfavorable formatting or insignificant experimental emphasis.
  4. We also excluded datasets with sample sizes that were less than our preselected cutoff sample size of three for each group.
  5. In addition, we have concentrated on a particular cell/tissue type in light of its influence on the course of a disease

Considering all the fact, we have found only one dataset for each disease.

  1. Also the authors should avoid comparing data from whole transcriptome analysis with data from micro-arrays.

>Answer: Thank you  for  your comments. Considering the criteria we have found only one micro-arrays dataset for moyamoya. Furthermore, previous study inspire us to continue the study with two different gene expression profiling methods [1, 2, 3]. 

  1. Auwul, M. R., Zhang, C., Rahman, M. R., Shahjaman, M., Alyami, S. A., & Moni, M. A. (2021). Network-based transcriptomic analysis identifies the genetic effect of COVID-19 to chronic kidney disease patients: A bioinformatics approach. Saudi journal of biological sciences28(10), 5647-5656.
  2. Rahman, M. H., Rana, H. K., Peng, S., Kibria, M. G., Islam, M. Z., Mahmud, S. H., & Moni, M. A. (2021). Bioinformatics and system biology approaches to identify pathophysiological impact of COVID-19 to the progression and severity of neurological diseases. Computers in biology and medicine138, 104859.
  3. Rahman, M. H., Rana, H. K., Peng, S., Hu, X., Chen, C., Quinn, J. M., & Moni, M. A. (2021). Bioinformatics and machine learning methodologies to identify the effects of central nervous system disorders on glioblastoma progression. Briefings in Bioinformatics22(5), bbaa365.

  1. explain why the number of genes in Raw Data from the glioblastoma dataset is so low compared to the other two diseases (Table 1). It is not clear in the text which data was retrieved from micro-arrays: Glioblastoma or moyamoya.

>Answer:  Thank you for your comments. We have modified and added the information in section 2.1 in our manuscript according your comment which is shown as highlighted.

3) provide details on how the analysis were performed (which versions of the algorithms and what settings were selected) and also upload the code generated in github

>Answer : Thank you for your comments. We have added the information in section 2.2 in our manuscript according your comment which is shown as highlighted.

4) evaluate whether some of the emerging targets are druggable

>Answer:  Thank you for your comments. We have added the section 2.6 and 3.5 in our manuscript according your comment which is shown as highlighted.

5) edit the text, with the aim to produce a more compact manuscript focusing more on the results obtained and their biological and clinical relevance.

>Answer:  Thank you for your comments. We have discussed about the results in the discussion section which is shown as highlighted.

6) revise several spelling, grammar and syntax errors (some of them even in the manuscript title)

>Answer:  Thank you for your comments. We have modified our manuscript according to your comments.

Reviewer 3 Report

Article is well written. Need some minor modification in grammar and spell check.

More deep discussion is needed 

Author Response

Article is well written. Need some minor modification in grammar and spell check.

More deep discussion is needed 

>Answer:  Thank you for your comments. We have modified our manuscript and discussed about the results in the discussion section which is shown as highlighted.

Round 2

Reviewer 1 Report

The authors have answered my comments and the paper is suitable for publication.

However, as the author reviewers have pointed out, the English language needs some polishing. 

In addition, the authors need to use the template of the journal and not rely on the fact that the editorial office will do their job.

Author Response

Reviewer – 01

The authors have answered my comments and the paper is suitable for publication.

However, as the author reviewers have pointed out, the English language needs some polishing. 

Answer: Thank you for your comments. We have modified our manuscript according to your comments.

In addition, the authors need to use the template of the journal and not rely on the fact that the editorial office will do their job.

Answer: Thank you for your observation. We have updated our manuscript in the specified journal’s template.

Reviewer 2 Report

The new version of the manuscript is much improved and the authors have addressed most of the points raised. I still think that language and style should be greatly improved prior to publication. Section 2.6, for example, should be better written, while several spelling errors persist (for example the first line of Section 4: "its" should be changed to "it is").

Author Response

The new version of the manuscript is much improved and the authors have addressed most of the points raised. I still think that language and style should be greatly improved prior to publication. Section 2.6, for example, should be better written, while several spelling errors persist (for example the first line of Section 4: "its" should be changed to "it is").

Answer: Thank you  for  your observation. We have made the changes and highlighted accordingly in our manuscript.